# Efficacy and safety of endocrine therapy after mastectomy in patients with hormone receptor positive breast ductal carcinoma in situ: Retrospective cohort study

Nan Niu[1†], Yinan Zhang[1†], Yang Bai[2†], Xin Wang[3], Shunchao Yan[1], Dong Song[4], Hong Xu[5], Tong Liu[6], Bin Hua[7], Yingchao Zhang[8], Jinchi Liu[1], Xinbo Qiao[1], Jiaxiang Liu[3], Xinyu Zheng[9]*, Hongyi Cao[10]*, Caigang Liu[1]*

[1]Department of Oncology, Shengjing Hospital of China Medical University, Shenyang, China; [2]Department of Nursing, Shengjing Hospital of China Medical University, Shenyang, China; [3]Department of Breast Surgery, National Cancer Center/National Clinical Research Center for Cancer/Cancer Hospital, Beijing, China; [4]Department of Breast Surgery, the First Hospital of Jilin University, Changchun, China; [5]Department of Breast Surgery, Liaoning Cancer Hospital and Institute, Shenyang, China; [6]Department of Breast Surgery, Cancer Hospital of Harbin Medical University, Harbin, China; [7]Department of General Surgery, Beijing Hospital, National Center of Gerontology, Institute of Geriatric Medicine, Beijing, China; [8]Department of Breast Surgery, the Second Hospital of Jilin University, Changchun, China; [9]Department of Breast Surgery, the First Affiliated Hospital of China Medical University, Shenyang, China; [10]Department of Pathology, the First Affiliated Hospital of China Medical University and College of Basic Medical Sciences, Shenyang, China

*For correspondence:
xyzheng@cmu.edu.cn (XZ);
caohongyi905@163.com (HC);
angel-s205@163.com (CL)

†These authors contributed equally to this work

## Abstract

**Background:** More than half of Chinese patients with hormone receptor positive (HR+) ductal carcinoma in situ (DCIS) are treated with mastectomy, and usually subjected to postoperative endocrine therapy (ET). Given that long-term ET can cause severe adverse effects it is important to determine the beneficial effect and safety of post-mastectomy ET on the disease-free survival (DFS) and adverse events in patients with HR+ DCIS.

**Methods:** To explore beneficial effect and safety of post-mastectomy ET in patients with HR+ DCIS, we performed a multicenter, population-based study. This retrospective study analyzed the DFS and adverse events in 1037 HR+ DCIS Chinese patients with or without post-mastectomy ET from eight breast centers between 2006 and 2016. The median follow-up time period was 86 months.

**Results:** There were 791 DCIS patients receiving ET (ET group). Those patients were followed up for a median of 86 months (range, 60–177 months). There were 23 cases with tumor recurrence or distant metastasis. There were similar 5-year DFS rates and DFS between the ET and non-ET groups, even for those with high-risk factors. Conversely, 37.04% of patients suffered from adverse events after ET, which were significantly higher than those in the non-ET group.

**Conclusions:** ET after mastectomy did not benefit patients with HR+ DCIS for their DFS, rather increased adverse events in those patients. Therefore, ET after mastectomy may not be

recommended for patients with HR+ DCIS, even for those with high-risk factors, such as multifocal, microinvasive, and higher T stage.

**Funding:** This study was supported by grants from Outstanding Scientific Fund of Shengjing Hospital (201803) and Outstanding Young Scholars of Liaoning Province (2019-YQ-10).

## Editor's evaluation

This valuable study describes the effects of endocrine therapy in a large series of Chinese patients treated with mastectomy (both efficacy and side effects). Whilst there are some caveats regarding the methodology (retrospective, numbers of events, and some potential methodological bias in data collection) this is an important piece of work and with further, ideally prospective, data collection, has the potential to markedly improve the management of patients with DCIS.

## Introduction

Breast cancer screening in adult women has improved its early detection, increasing incidence of ductal carcinoma in situ (DCIS), which currently accounts for>20% of all new breast cancers in USA (*Ward et al., 2015*; *Siegel et al., 2021*). Breast-conserving surgery (BCS) plus radiotherapy (RT) has been widely used for the control of invasive cancer recurrence (*Shah et al., 2016*). Recent studies have shown that there is an increase in the percentages of DCIS patients for unilateral and bilateral mastectomy in USA, particularly for young patients (*Byun et al., 2021*). There are approximately 30% of DCIS patients receiving mastectomy and potential breast reconstruction, especially for those with widespread, multicentric DCIS in USA (*Wärnberg et al., 2014*; *Worni et al., 2015*). However, there are nearly 60% of DCIS patients receiving mastectomy in China, particularly in the economic underdeveloped regions, because they have a fear of cancer recurrence (FCR) and worry subsequent treatment costs.

Endocrine therapy (ET) with tamoxifen or aromatase inhibitor (AI) (letrozole, anastrozole, exemestane) has been recommended for hormone receptor positive (HR+) breast cancer patients after BCS plus RT to reduce the risk of contralateral breast cancer (CBC) and ipsilateral breast tumor recurrence by National Comprehensive Cancer Network (*Allred et al., 2012*; *Forbes et al., 2016*; *Ganz et al., 2016*). Premenopausal or perimenopausal patients also receive a subcutaneous injection with goserelin. It is notable that ET after bilateral mastectomy is not recommended for HR+ DCIS patients, who have a minimal risk for disease recurrence. However, ET is still being used for some HR+ DCIS patients post unilateral mastectomy in Western countries because ET has been thought to reduce the risk of contralateral recurrence of invasive and pure DCIS (*Byun et al., 2021*). In China, ET has been widely used for HR+ DCIS patients after mastectomy because of FCR although no specific recommendation of ET for them (*Mao et al., 2021*). Moreover, long-term ET can cause adverse effects, particularly for postmenopausal women. However, there is no report on whether ET after unilateral mastectomy can benefit Chinese HR+ DCIS patients for reducing contralateral recurrence of breast cancer and prolonging disease-free survival (DFS) as well as its safety. Accordingly, this retrospective cohort study evaluated the efficacy and safety of ET after mastectomy in the DFS and adverse events of 1037 HR+ DCIS patients.

## Methods
### Subjects

This study was approved by the Institutional Review Board of Shengjing Hospital (approval number: 2020PS014K). This retrospective cohort study reviewed and analyzed the DFS and adverse events in 1037 HR+ DCIS patients with, or without, ET after mastectomy from December 2006 to August 2016 (*Figure 1*). The inclusion criteria included: (1) age >18; (2) pathological diagnosis of estrogen receptor low-positive (1–10% nuclei staining) and positive (>10% nuclei staining, using methodology outlined

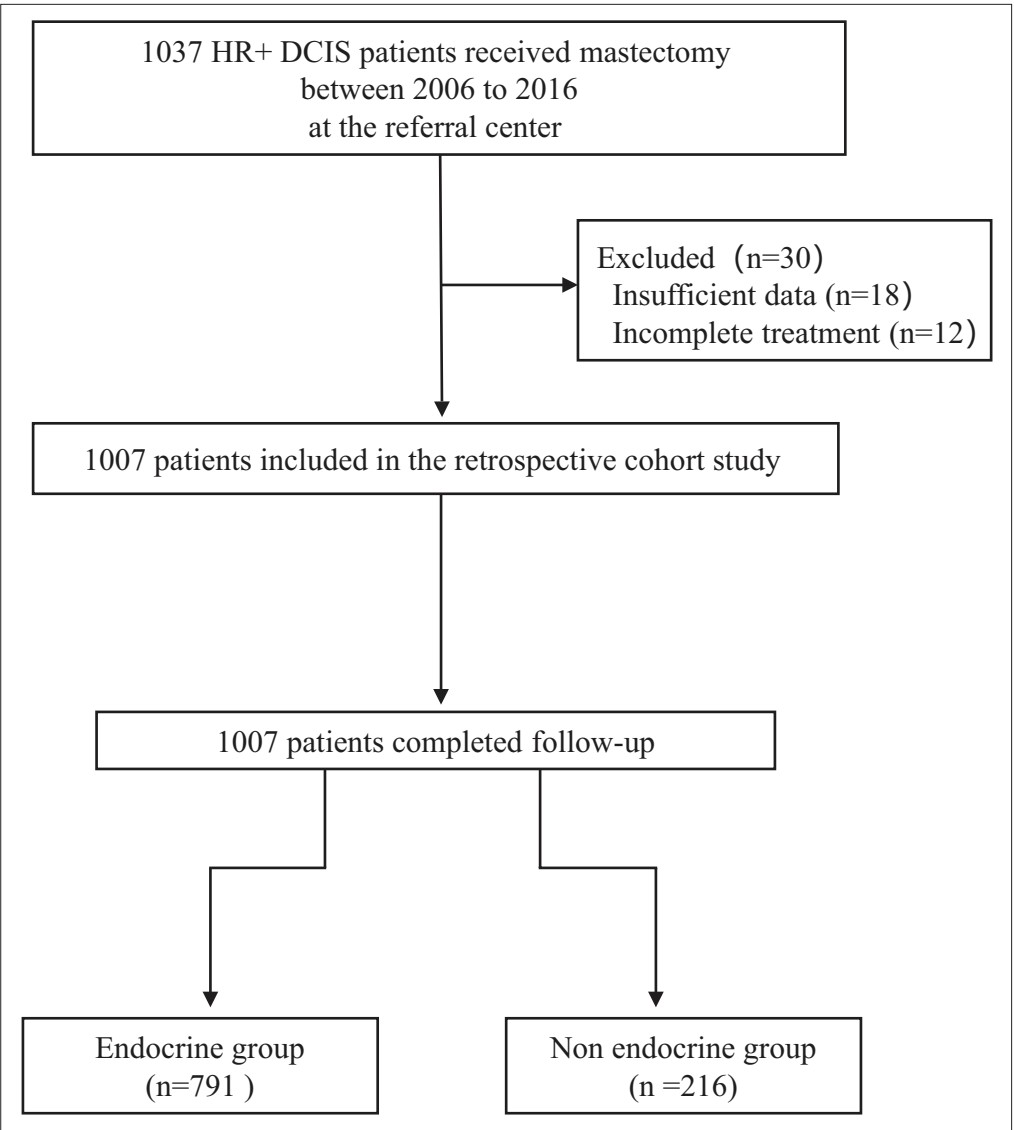

**Figure 1.** Flowchart of the study.

in the ASCO/CAP HR testing guideline) DCIS regardless of progesterone receptor expression and Her-2 status; (3) receiving mastectomy regardless of treatment with ET; (4) complete medical records with regular postoperative follow-up for at least for 5 years.

## Data collection and statistical analysis

The demographic and clinical data, including age, menopausal status, diagnosis, tumor pathological index, adjuvant treatments (drugs, duration), the ET-related adverse effects, tumor recurrence, and survival status were collected by reviewing the case notes and follow-up records. The tumor recurrence was defined as pathologically confirmed breast cancer (DCIS, invasive breast cancer) or metastatic cancers. The DFS was calculated from the diagnosis to the tumor recurrence, or the last follow-up.

All patients were stratified, based on ET, and their demographic and clinical data were analyzed by Chi-squared, Fisher's exact, and Wilcoxon rank sum tests where applicable. Their DFS was estimated using Kaplan-Meier method and analyzed by the log-rank test. The potential risk of individual factors

**Table 1.** The demographic and clinical characteristics of patients.

| | ET (N=791) | Non-ET (N=216) | p-Value |
|---|---|---|---|
| Age (n, %) | | | 0.134 |
| ≤50 | 448 (57%) | 110 (51%) | |
| >50 | 343 (43%) | 106 (49%) | |
| Tumor size (n, %) | | | 0.839 |
| ≤20 mm | 459 (58%) | 127 (59%) | |
| >20 mm | 332 (42%) | 89 (41%) | |
| Microinvasive (n, %) | | | 0.322 |
| Yes | 128 (16%) | 29 (13%) | |
| No | 663 (84%) | 187 (87%) | |
| Tumor grade (n, %) | | | 0.190 |
| I-II | 565 (71%) | 164 (76%) | |
| III | 226 (29%) | 52 (24%) | |
| Ki67 (n, %) | | | 0.071 |
| ≥15% | 279 (35%) | 62 (29%) | |
| <15% | 512 (65%) | 154 (71%) | |
| Multifocal (n, %) | | | 0.310 |
| Yes | 64 (8%) | 13 (6%) | |
| No | 727 (92%) | 203 (94%) | |

Notes: Data are n (%). ET, endocrine therapy. Source files available in **Table 1—source data 1**.

The online version of this article includes the following source data for table 1:

**Source data 1.** The demographic and clinical characteristics of patients.

for the tumor recurrence was analyzed by hazard ratios and 95% confidence intervals. All statistical analyses were performed by SPSS 24.0 software (SPSS, Chicago, IL, USA). Statistical significance was defined when a two-tailed p-value is <0.05.

## Results

### ET after mastectomy does not significantly alter the DFS of HR+ DCIS patients

A total of 1007 HR+ DCIS patients with mastectomy were selected and their demographic and clinical data are shown in *Table 1*. According to ET treatment, those patients were stratified in the ET (n=791) and non-ET (n=216) groups. There was no significant difference in any of the demographic and clinical measures tested between these two groups. There were 23 cases (19 vs. 4 between the ET and non-ET groups) with tumor recurrence, leading to 2.40% vs. 1.85% (p>0.05, determined by Fisher's exact test) of tumor recurrence rate in the ET and non-ET groups throughout the 12-year post-surgical observation (*Table 2*). There were 4 cases with invasive local recurrence, 3 with CBC, 12 with distant metastasis in the ET group while 4 cases with distant metastases in the non-ET group. Stratification analyses indicated that the tumor recurrence rate was not significantly associated with these measures, including high-risk factors in this population and surgery methods (unilateral mastectomy or bilateral mastectomy) (p>0.05 for all, *Table 3*).

There was no significant difference in the percentages of patients with a 5-year DFS rate of 98.36% vs. 99.07% between the ET and non-ET groups (p=0.44, *Figure 2A*). Further analysis revealed that

**Table 2.** Tumor recurrence rates in patients with HR+ DCIS after mastectomy.

| Tumor recurrence | ET (N=19) | Non-ET (N=4) |
|---|---|---|
| Invasive local recurrence | 4 (21%) | 0 (0%) |
| Contralateral breast cancer | 3 (16%) | 0 (0%) |
| Distant metastasis | | |
| Bone | 6 (32%) | 1 (25%) |
| Liver | 2 (11%) | 2 (50%) |
| Lung | 1 (5%) | 0 (0%) |
| Brain | 1 (5%) | 0 (0%) |
| Abdominal cavity | 1 (5%) | 1 (25%) |
| Lymph nodes | 1 (5%) | 0 (0%) |

Notes: Data are n (%). HR, hormone receptor; DCIS, ductal carcinoma in situ; ET, endocrine therapy.
Source files available in **Table 2—source data 1**.

The online version of this article includes the following source data for table 2:

**Source data 1.** Tumor recurrences in patients with hormone receptor positive (HR+) ductal carcinoma in situ (DCIS) after mastectomy.

there was also no significant difference in the percentages of patients with DFS between these subgroups (p>0.05 for all, *Figure 2B–F*), suggesting that the age, larger tumor size, positive microinvasive, higher tumor grade, and Ki67 levels were not associated with increased risk of worse DFS in this population. Hence, ET after mastectomy did not significantly reduce the tumor recurrence rate and prolong the DFS in HR+ DCIS patients.

## The ET-related adverse effects

ET can cause musculoskeletal dysfunction, vasomotor symptoms, gynecological events, cardiovascular events, and abnormal liver function in women, particularly in postmenopausal women. There were 551 patients receiving tamoxifen, 223 with AI, and others beginning with tamoxifen and later switching to AI. Analysis of adverse events in both groups revealed that 37.04% of patients in the ET group developed at least one adverse event, including bone fracture or endometrial cancer (n=4 each), while 15.28% of cases with these events in the non-ET group. There were 14.54% of patients with musculoskeletal dysfunctions, such as arthralgia, joint stiffness, osteoporosis, or myalgia in the ET group and the percentages of some adverse events tested in the ET group were significantly higher than that in the non-ET group in this population (*Figure 3*). Adverse events between tamoxifen and AI in the ET group are shown in *Table 4*.

Therefore, ET after mastectomy was associated with increased risk for development of different types of adverse effects in patients with HR+ DCIS.

## Discussion

Therapeutic strategies for HR+ DCIS, including mastectomy or BCS plus RT, have achieved a similarly high survival rate in patients (*Mannu et al., 2020*; *Narod et al., 2015*). Although ET after BCS plus RT is recommended for patients with HR+ DCIS, and benefits for those with positive surgical margin (*Allred et al., 2012*; *Forbes et al., 2016*; *Ganz et al., 2016*; *Wapnir et al., 2011*), many Asian HR+ DCIS patients chose mastectomy and received ET (*Mao et al., 2021*; *Worni et al., 2015*). In the present study, 78.55% of HR+ DCIS patients were treated with post-mastectomy ET. More importantly, we found that there was no significant difference in a 5-year DFS rate and tumor recurrence rate in HR+ DCIS patients regardless of ET, even in those with high-risk factors for tumor recurrence. The proportion with microinvasive was 16% in the ET group, and 13% in the non-ET group, respectively. Further analysis indicated that patients

**Table 3.** Stratification analysis of tumor recurrence rates in patients with HR+ DCIS after mastectomy.

| Characteristic | ET (N=791) | Non-ET (N=216) | HR (95% CI) | p-Value |
|---|---|---|---|---|
| Total | 19 (791) | 4 (216) | 1.30 (0.48–3.52) | 0.64 |
| Age | | | | |
| ≤50 | 12 (448) | 1 (110) | 2.91 (0.74–11.47) | 0.28 |
| >50 | 7 (343) | 3 (106) | 0.75 (0.18–3.17) | 0.67 |
| Tumor size | | | | |
| ≤20 mm | 6 (459) | 2 (127) | 0.82 (0.15–4.44) | 0.81 |
| >20 mm | 13 (332) | 2 (89) | 1.75 (0.51–6.04) | 0.45 |
| Microinvasive | | | | |
| Yes | 8 (128) | 0 (29) | 3.48 (0.60–20.02) | 0.16 |
| No | 11 (663) | 4 (187) | 0.76 (0.22–2.59) | 0.64 |
| Tumor grade | | | | |
| I-II | 10 (565) | 0 (164) | 3.64 (0.82–16.06) | 0.09 |
| III | 9 (226) | 4 (52) | 0.51 (0.13–2.07) | 0.26 |
| ER | | | | |
| 1–10% | 4 (165) | 2 (40) | 0.47 (0.08–2.67) | 0.386 |
| >10% | 15 (626) | 2 (176) | 2.14 (0.48–9.43) | 0.305 |
| Ki67 | | | | |
| ≥15% | 7 (279) | 1 (62) | 0.88 (0.09–8.29) | 0.67 |
| <15% | 12 (512) | 3 (154) | 1.42 (0.46–4.38) | 0.74 |
| Multifocal | | | | |
| Yes | 3 (64) | 0 (13) | 3.34 (0.17–67.46) | 0.43 |
| No | 16 (727) | 4 (203) | 1.11 (0.38–3.22) | 0.85 |
| Surgery | | | | |
| Unilateral mastectomy | 19 (776) | 4 (215) | 1.32 (0.45–3.93) | 0.61 |
| Bilateral mastectomy | 0 (15) | 0 (1) | – | – |

Notes: HR, hormone receptor; DCIS, ductal carcinoma in situ; ET, endocrine therapy.
Source files available in *Table 3—source data 1*.

The online version of this article includes the following source data for table 3:

**Source data 1.** Tumor recurrences in subgroups.

with high grade of DCIS in the microinvasive subgroup accounted for 42% and 10% in the ET group and non-ET group, respectively. This might explain why the DFS of those with microinvasive DCIS in the ET group trended to be worse than those in the non-ET group although there was no significant difference between them.

To the best of our knowledge, this was the first report on the efficacy of ET after mastectomy in the DFS of Chinese HR+ DCIS patients and these novel findings clearly indicated that ET after mastectomy did not prolong the DFS of HR+ DCIS patients.

Long-term ET can cause multiple adverse effects, affecting the life quality of patients. Indeed, 37.04% of patients developed adverse events following ET. Quantitative analysis revealed that the percentages of patients with musculoskeletal dysfunction, gynecological events, and abnormal liver function, but not vasomotor symptoms and cardiovascular events, in the ET group were significantly higher than that in the non-ET group of patients. The increased percentages of patients with

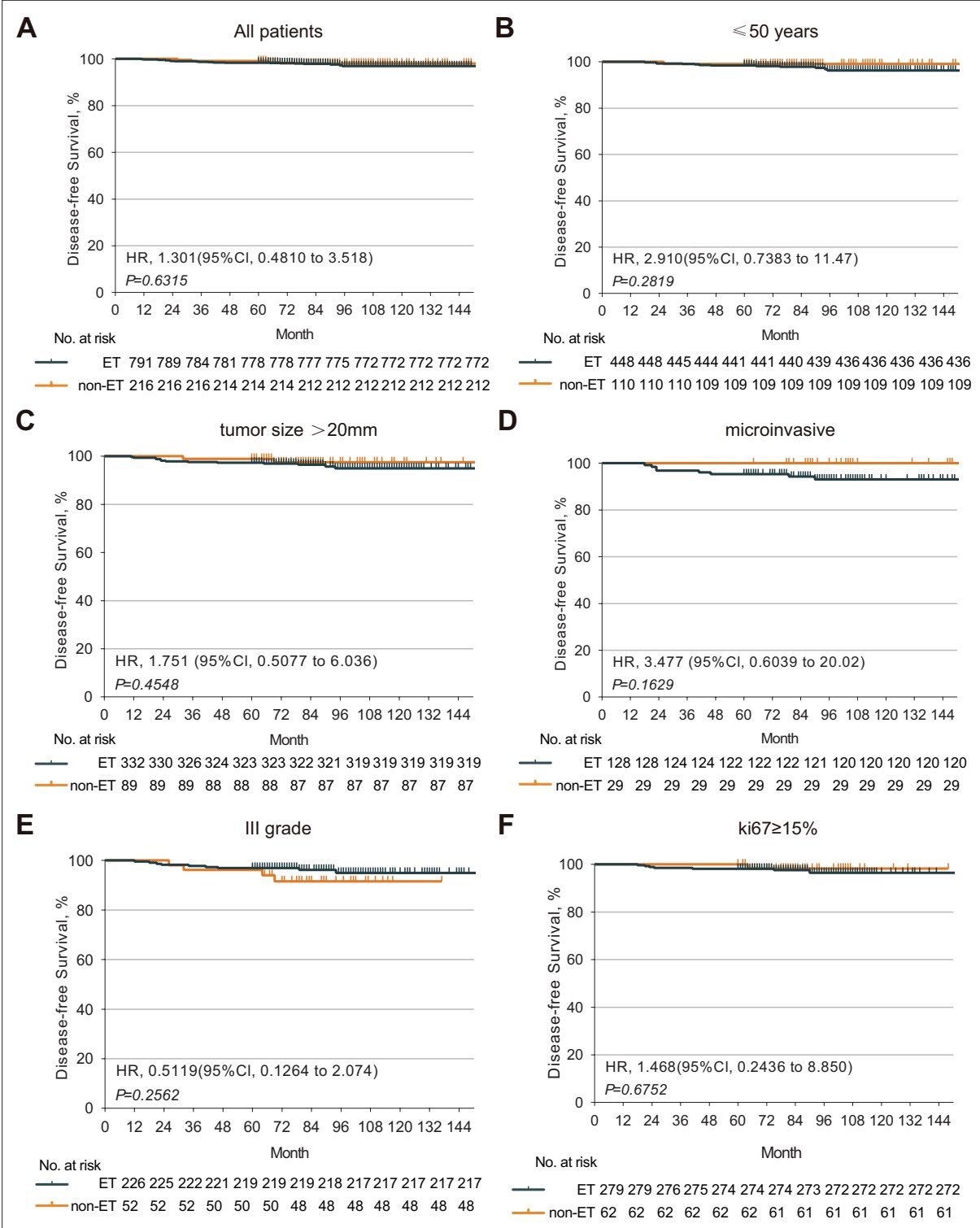

**Figure 2.** The DFS of HR+ DCIS patients with or without post-mastectomy ET. Kaplan-Meier analysis indicated that there was no significant difference in the DFS of HR+ DCIS patients between those with and without post-mastectomy ET. (**A**) There was no significant difference in the DFS of HR+ DCIS patients with age <50, (**B**) a larger tumor, (**C**) positive microinvasive, (**D**) higher tumor grade, (**E**) higher Ki67 level, (**F**) between those with and

*Figure 2 continued on next page*

*Figure 2 continued*

without post-mastectomy ET. HR, hormone receptor; DCIS, ductal carcinoma in situ; ET, endocrine therapy; DFS, disease-free survival. Source files available in *Figure 2—source data 1*.

The online version of this article includes the following source data for figure 2:

**Source data 1.** The disease-free survival of hormone receptor positive (HR+) ductal carcinoma in situ (DCIS) patients between endocrine therapy (ET) and non-ET groups.

these clinical symptoms demonstrated that long-term ET caused multiple adverse effects in HR+ DCIS patients after mastectomy. Given that the majority of HR+ DCIS patients chose mastectomy with a long DFS and ET after mastectomy did not prolong their DFS, rather significantly increased ET-related adverse effects in those patients, our findings suggest that ET may be decreased for its dose and duration or completely avoided for HR+ DCIS patients following mastectomy to improve their life quality.

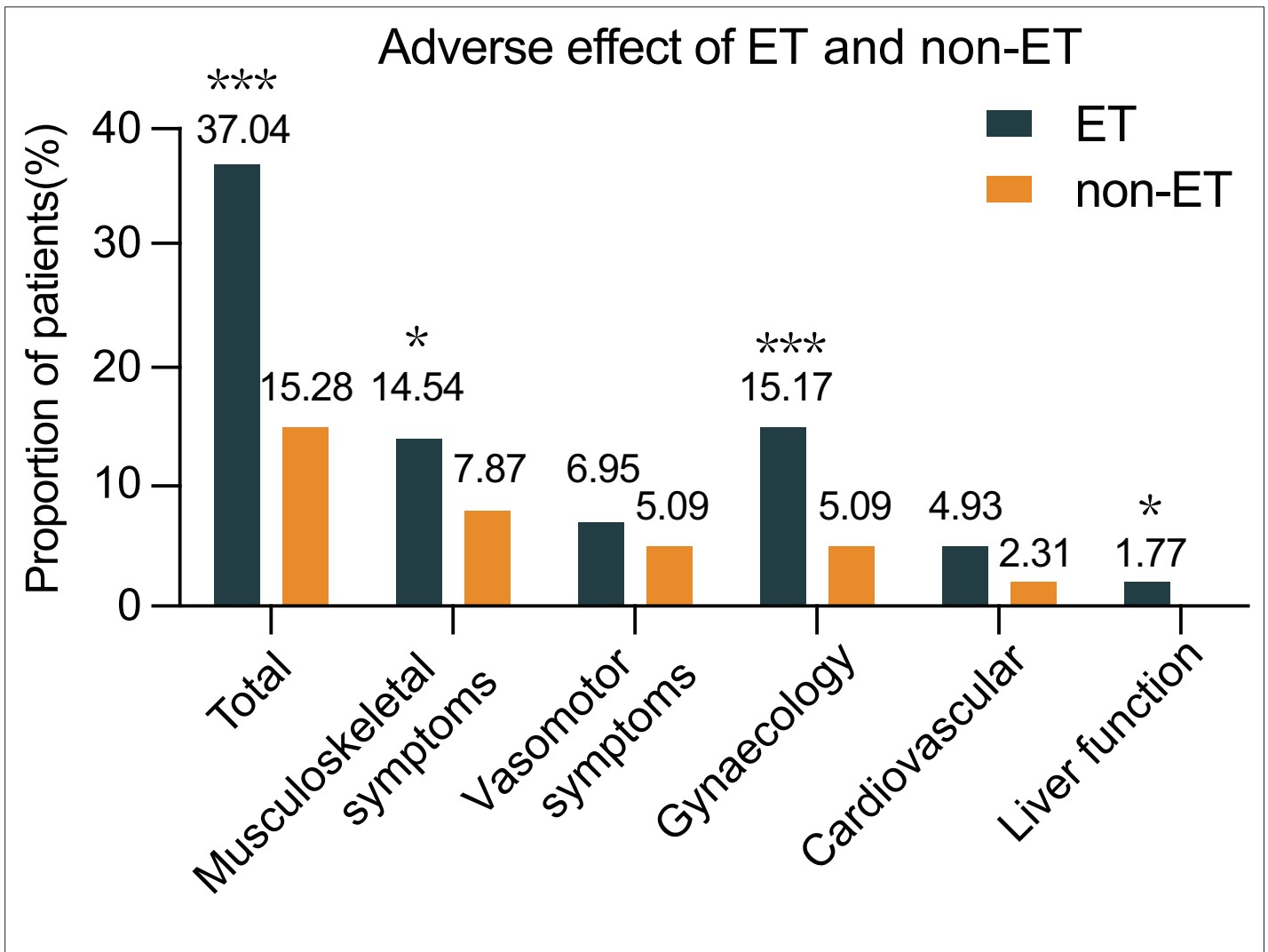

**Figure 3.** The frequency of patients with adverse effects between the ET and non-ET groups. Data are expressed as % of cases with adverse events and real case numbers labeled and analyzed by Chi-squared test. *p<0.05, ***p<0.001. ET, endocrine therapy. Source files available in *Figure 3—source data 1*.

The online version of this article includes the following source data for figure 3:

**Source data 1.** Adverse events between the endocrine therapy (ET) and non-ET groups.

**Table 4.** Adverse events between TAM and AI in the ET group.

| Adverse events | TAM (N=551) | AI (N=223) |
|---|---|---|
| Total | 212 (38%) | 76 (34%) |
| Musculoskeletal symptoms | 61 (11%) | 52 (23%) |
| Vasomotor symptoms | 42 (8%) | 12 (5%) |
| Gynecological events | 113 (21%) | 5 (2%) |
| Cardiovascular events | 26 (5%) | 12 (5%) |
| Abnormal liver function | 10 (2%) | 4 (2%) |

Notes: TAM, tamoxifen; AI, aromatase inhibitor; ET, endocrine therapy.
Source files available in *Table 4—source data 1*.

The online version of this article includes the following source data for table 4:

**Source data 1.** Adverse events in the endocrine therapy (ET) group.

We recognized that our study was retrospective in nature and with a small number of events. Further validation of these findings is warranted in a large, randomized prospective trial to better understand the benefits and toxicities of ET in HR+ DCIS patients and inform future care.

## Conclusions

ET after mastectomy did not prolong the DFS of HR+ DCIS patients, rather increased adverse effects. Therefore, ET may be decreased for its dose and duration or completely avoided for HR+ DCIS patients after mastectomy.

## Additional information

### Competing interests

Caigang Liu: Senior editor, *eLife*. The other authors declare that no competing interests exist.

### Funding

| Funder | Grant reference number | Author |
|---|---|---|
| Outstanding Young Scholars of Liaoning Province | 2019-YQ-10 | Caigang Liu |
| Outstanding Scientific Fund of Shengjing Hospital | 201803 | Caigang Liu |

The funders had no role in study design, data collection and interpretation, or the decision to submit the work for publication.

### Author contributions

Nan Niu, Resources, Formal analysis, Writing – original draft, Writing - review and editing; Yinan Zhang, Resources, Data curation, Formal analysis; Yang Bai, Xin Wang, Dong Song, Hong Xu, Tong Liu, Bin Hua, Yingchao Zhang, Jiaxiang Liu, Resources, Data curation; Shunchao Yan, Formal analysis, Writing – original draft; Jinchi Liu, Xinbo Qiao, Formal analysis, Methodology; Xinyu Zheng, Conceptualization, Supervision, Project administration; Hongyi Cao, Resources, Data curation, Formal analysis, Investigation; Caigang Liu, Conceptualization, Supervision, Funding acquisition, Validation, Project administration, Writing - review and editing

### Author ORCIDs

Nan Niu ⓘ http://orcid.org/0000-0002-8206-941X
Xinbo Qiao ⓘ http://orcid.org/0000-0002-6759-921X
Caigang Liu ⓘ http://orcid.org/0000-0003-2083-235X

### Ethics

Human subjects: This study was approved by the Institutional Review Board of Shengjing Hospital (approval number: 2020PS014K) and adhered to the principles outlined in the Declaration of Helsinki and Good Clinical Practice. Written informed consent was obtained from all patients.

### Decision letter and Author response

Decision letter https://doi.org/10.7554/eLife.83045.sa1
Author response https://doi.org/10.7554/eLife.83045.sa2

---

## Additional files

### Supplementary files

- MDAR checklist
- Reporting standard 1. STROBE checklist.

### Data availability

All data generated or analyzed during this study are included in the manuscript and supporting files.

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
