## [Editor Report]

This valuable study describes the effects of endocrine therapy in a large series of Chinese patients treated with mastectomy (both efficacy and side effects). Whilst there are some caveats regarding the methodology (retrospective, numbers of events, and some potential methodological bias in data collection) this is an important piece of work and with further, ideally prospective, data collection, has the potential to markedly improve the management of patients with DCIS.

---

## [Decision Letter]

**Decision letter after peer review:**

Thank you for submitting your article "Efficacy and safety of endocrine therapy after mastectomy in patients with hormone receptor positive breast ductal carcinoma in situ: retrospective cohort study" for consideration by *eLife*. Your article has been reviewed by 4 peer reviewers, including Sarah Pinder as Reviewing Editor and Reviewer #1, and the evaluation has been overseen by Tony Ng as the Senior Editor. The following individuals involved in the review of your submission have agreed to reveal their identity: David Dodwell (Reviewer #3); Manish Charan (Reviewer #4).

Essential revisions:

1) Detail on the definition and assessment/cut-points for defining hormone receptor positivity should be included.

2) Comment on the proportion of grades of DCIS would be appropriate, as the distribution is not what one might expect in international series.

3) The authors should assess what the analysis shows when HR+ DCIS patients with ET after mastectomy are divided into a unilateral mastectomy group vs bilateral mastectomy group to assess whether the effect of ET is different.

4) The methodology for collecting information on side effects should be described.

5) The authors should emphasize that this study is retrospective in nature, with a small number of events, and are encouraged to consider (or at least outline the need for) a large, randomized trial in DCIS to better understand the benefits and toxicities of endocrine therapy in DCIS and thus better inform future care.

*Reviewer #1 (Recommendations for the authors):*

More detail on the definition and assessment/cut-points for defining hormone receptor positivity should be included. At present, it is not clear if this means oestrogen or/and progesterone receptor, how this has been assessed, and with what cut-point to define 'positive'. I am not certain if it would be possible to include the degree of positivity in analyses.

Comment on the proportion of grades of DCIS would be appropriate, as the distribution is not what one might expect and it is not clear why this is so different from what one sees in population series, for example from the UK.

Similarly, based on the grades reported, the proportion with microinvasive disease is high and the disease-free survival of those with microinvasive disease and in receipt of ET does (non-significantly) seem to show a trend to poorer DFS (Figure 2, D). This might be worthy of further comment.

*Reviewer #2 (Recommendations for the authors):*

The manuscript by Niu and colleagues, entitled "Efficacy and safety of endocrine therapy after mastectomy in patients with hormone receptor positive breast ductal carcinoma in situ: retrospective cohort study." reported that ET after mastectomy did not prolong the DFS of Chinese HR+ DCIS patients, rather increased adverse effects. For the first time, the authors analyzed beneficial effect and safety of ET after mastectomy in Chinese patients with HR+ DCIS through the clinical case review. The study had a large number of cases, a long follow-up period, a large workload, and a relatively simple research method, which made the results reliable. The conclusion of this study is of great significance to guide the choice of appropriate treatment for Chinese patients with HR+ DCIS, and it has obvious benefits to reduce the economic burden of the patient's family and improve the quality of life for patients. However, there is a question that needs to be considered by the authors: Whether HR+ DCIS patients with ET after mastectomy are divided into unilateral mastectomy group and bilateral mastectomy group to calculate the effect of ET is significantly different.

*Reviewer #3 (Recommendations for the authors):*

I would encourage the authors to consider conducting a large, randomized trial in DCIS to better understand the benefits and toxicities of endocrine therapy in DCIS and thus better inform future care.

---

## [Author Response]

Essential revisions:1) Detail on the definition and assessment/cut-points for defining hormone receptor positivity should be included.

We are sorry for the confusion. The inclusion criteria included pathological diagnosis of estrogen receptor low-positive (1%-10% nuclei staining) and positive (>10% nuclei staining, using methodology outlined in the ASCO/CAP HR testing guideline) DCIS regardless of progesterone receptor expression. We have added the detailed information on the definition and assessment for defining hormone receptor positivity in the Methods section of the revision (line 96-98). Further stratification analyses indicated the tumor recurrence rate was no significant difference between the ER-low-positive and ER-positive subgroups (P>0.05, Table 3).

Reference:

1. Allison Kimberly H, Hammond M Elizabeth H, Dowsett Mitchell et al. Estrogen and Progesterone Receptor Testing in Breast Cancer: ASCO/CAP Guideline Update. [J]. J Clin Oncol, 2020, 38: 1346-1366.

2) Comment on the proportion of grades of DCIS would be appropriate, as the distribution is not what one might expect in international series.

In fact, clinical trials have reported different proportions of grades of DCIS. While 60% of all cases were of high histological grade in UK ^1^, the proportion of high histological grade was 34% (206/601) in the Tamoxifen group and 32% (192/592) in the Anastrozole group in the NSABP B-35 study ^2^. Furthermore, high histological grade of DCIS accounted for 39% (587/1489) in the Tamoxifen group and 37% (542/1449) in the Anastrozole group in the IBIS-II DCIS trial ^3^. A study of Chinese DCIS patients also reported that the percentage of high histological grade of DCIS was 21.9% (135/617) ^4^. The proportion of cases with high-grade DCIS in our study (29% vs 24%, respectively) was similarly to other reports.

Reference:

1. Mannu, G. S., Wang, Z., Broggio, J., Charman, J., Cheung, S., Kearins, O.,... Darby, S. C. (2020). Invasive breast cancer and breast cancer mortality after ductal carcinoma in situ in women attending for breast screening in England, 1988-2014: population based observational cohort study. *BMJ, 369*, m1570. doi:10.1136/bmj.m1570

2. Ganz, P. A., Cecchini, R. S., Julian, T. B., Margolese, R. G., Costantino, J. P., Vallow, L. A.,... Wolmark, N. (2016). Patient-reported outcomes with anastrozole versus tamoxifen for postmenopausal patients with ductal carcinoma in situ treated with lumpectomy plus radiotherapy (NSABP B-35): a randomised, double-blind, phase 3 clinical trial. The Lancet, 387(10021), 857-865. doi:10.1016/s0140-6736(15)01169-1

3. Forbes, J. F., Sestak, I., Howell, A., Bonanni, B., Bundred, N., Levy, C.,... Cuzick, J. (2016). Anastrozole versus tamoxifen for the prevention of locoregional and contralateral breast cancer in postmenopausal women with locally excised ductal carcinoma in situ (IBIS-II DCIS): a double-blind, randomised controlled trial. *The Lancet, 387*(10021), 866-873. doi:10.1016/s0140-6736(15)01129-0

4. Mao, G., Shi, X., Wang, X., Zhang, X., Chen, X., Ma, J.,... Guo, X. (2021). Clinicopathological Characteristics of Breast Ductal Carcinoma In Situ: An Analysis of Chinese Population of 617 Patients. Journal of oncology, 2021, 8854418. doi:10.1155/2021/8854418

3) The authors should assess what the analysis shows when HR+ DCIS patients with ET after mastectomy are divided into a unilateral mastectomy group vs bilateral mastectomy group to assess whether the effect of ET is different.

There were only sixteen patients, who received bilateral mastectomy. Among the sixteen patients, fifteen received ET and one patient with tumor recurrence in the ET group. Further analyses indicated that the tumor recurrence rate was not significantly different between the unilateral mastectomy group and bilateral mastectomy group [P = 0.54, HR 0.97 (0.95 to 0.98)]. We have added it in the Results section of the revision (line 125-126).

4) The methodology for collecting information on side effects should be described.

The adverse effects were gathered through reviewing the case notes and follow-up records. We have added it in the Methods section of the revision (line 104).

5) The authors should emphasize that this study is retrospective in nature, with a small number of events, and are encouraged to consider (or at least outline the need for) a large, randomized trial in DCIS to better understand the benefits and toxicities of endocrine therapy in DCIS and thus better inform future care.

We recognized that our study was retrospective in nature and with a small number of events. Further validation of these findings is warranted in a large, randomized trial in DCIS to better understand the benefits and toxicities of endocrine therapy in DCIS and better inform future care. In response to his/her concerns, we have discussed the limitation in the revision (line 176-179).

Reviewer #1 (Recommendations for the authors):More detail on the definition and assessment/cut-points for defining hormone receptor positivity should be included. At present, it is not clear if this means oestrogen or/and progesterone receptor, how this has been assessed, and with what cut-point to define 'positive'. I am not certain if it would be possible to include the degree of positivity in analyses.

We are sorry for the confusion. The inclusion criteria included pathological diagnosis of estrogen receptor low-positive (1%-10% nuclei staining) and positive (>10% nuclei staining, using methodology outlined in the ASCO/CAP HR testing guideline) DCIS regardless of progesterone receptor expression. We have added the detail information on the definition and assessment for defining hormone receptor positivity in the Methods section of the revision (line 96-98). Further stratification analyses indicated the tumor recurrence rate was not significantly different between the ER-low-positive and ER-positive subgroups (P>0.05, Table 3).

Reference:

1. Allison Kimberly H,Hammond M Elizabeth H,Dowsett Mitchell et al. Estrogen and Progesterone Receptor Testing in Breast Cancer: ASCO/CAP Guideline Update.[J].J Clin Oncol, 2020, 38: 1346-1366.

Comment on the proportion of grades of DCIS would be appropriate, as the distribution is not what one might expect and it is not clear why this is so different from what one sees in population series, for example from the UK.

In fact, clinical trials have reported different proportions of grades of DCIS. While 60% of all cases were of high histological grade in UK ^1^, the proportion of high histological grade was 34% (206/601) in the Tamoxifen group and 32% (192/592) in the Anastrozole group in the NSABP B-35 study ^2^. Furthermore, high histological grade of DCIS accounted for 39% (587/1489) in the Tamoxifen group and 37% (542/1449) in the Anastrozole group in the IBIS-II DCIS trial^3^. A study of Chinese DCIS patients also reported that the percentage of high histological grade of DCIS was 21.9% (135/617) ^4^. The proportion of cases with high-grade DCIS in our study (29% vs 24%, respectively) was similarly to other reports.

Reference:

1. Mannu, G. S., Wang, Z., Broggio, J., Charman, J., Cheung, S., Kearins, O.,... Darby, S. C. (2020). Invasive breast cancer and breast cancer mortality after ductal carcinoma in situ in women attending for breast screening in England, 1988-2014: population based observational cohort study. BMJ, 369, m1570. doi:10.1136/bmj.m1570

2. Ganz, P. A., Cecchini, R. S., Julian, T. B., Margolese, R. G., Costantino, J. P., Vallow, L. A.,... Wolmark, N. (2016). Patient-reported outcomes with anastrozole versus tamoxifen for postmenopausal patients with ductal carcinoma in situ treated with lumpectomy plus radiotherapy (NSABP B-35): a randomised, double-blind, phase 3 clinical trial. The Lancet, 387(10021), 857-865. doi:10.1016/s0140-6736(15)01169-1

3. Forbes, J. F., Sestak, I., Howell, A., Bonanni, B., Bundred, N., Levy, C.,... Cuzick, J. (2016). Anastrozole versus tamoxifen for the prevention of locoregional and contralateral breast cancer in postmenopausal women with locally excised ductal carcinoma in situ (IBIS-II DCIS): a double-blind, randomised controlled trial. The Lancet, 387(10021), 866-873. doi:10.1016/s0140-6736(15)01129-0

4. Mao, G., Shi, X., Wang, X., Zhang, X., Chen, X., Ma, J.,... Guo, X. (2021). Clinicopathological Characteristics of Breast Ductal Carcinoma In Situ: An Analysis of Chinese Population of 617 Patients. Journal of oncology, 2021, 8854418. doi:10.1155/2021/8854418

Similarly, based on the grades reported, the proportion with microinvasive disease is high and the disease-free survival of those with microinvasive disease and in receipt of ET does (non-significantly) seem to show a trend to poorer DFS (Figure 2, D). This might be worthy of further comment.

The proportion with microinvasive DCIS was 16% in the ET group, and 13% in the non-ET group, respectively. Further analysis indicated that patients with high grade of DCIS in the microinvasive subgroup accounted for 42% and 10% in the ET group and non-ET group, respectively. This might explain why the DFS of those with microinvasive DCIS in the ET group trended to be worse than those in the non-ET group although there was no significant difference between them. We have discussed this point in the section of Discussion (line 156-161).

Reviewer #2 (Recommendations for the authors):The manuscript by Niu and colleagues, entitled "Efficacy and safety of endocrine therapy after mastectomy in patients with hormone receptor positive breast ductal carcinoma in situ: retrospective cohort study." reported that ET after mastectomy did not prolong the DFS of Chinese HR+ DCIS patients, rather increased adverse effects. For the first time, the authors analyzed beneficial effect and safety of ET after mastectomy in Chinese patients with HR+ DCIS through the clinical case review. The study had a large number of cases, a long follow-up period, a large workload, and a relatively simple research method, which made the results reliable. The conclusion of this study is of great significance to guide the choice of appropriate treatment for Chinese patients with HR+ DCIS, and it has obvious benefits to reduce the economic burden of the patient's family and improve the quality of life for patients. However, there is a question that needs to be considered by the authors: Whether HR+ DCIS patients with ET after mastectomy are divided into unilateral mastectomy group and bilateral mastectomy group to calculate the effect of ET is significantly different.

There were only sixteen patients, who received bilateral mastectomy. Among the sixteen patients, fifteen received ET and one patient with tumor recurrence in the ET group. Further analyses indicated that the tumor recurrence rate was not significantly different between the unilateral mastectomy group and bilateral mastectomy group [P = 0.54, HR 0.97 (0.95 to 0.98)]. We have added it in the Results section of the revision (line 125-126).

Reviewer #3 (Recommendations for the authors):I would encourage the authors to consider conducting a large, randomized trial in DCIS to better understand the benefits and toxicities of endocrine therapy in DCIS and thus better inform future care.

We recognized that our study was retrospective in nature and with a small number of events. Further validation of these findings is warranted in a large, randomized trial in DCIS to better understand the benefits and toxicities of endocrine therapy in DCIS and better inform future care. In response to his/her concerns, we have discussed the limitations in the revision (line 176-179).